# Hemorheological and Microcirculatory Relations of Acute Pancreatitis

**DOI:** 10.3390/metabo13010004

**Published:** 2022-12-20

**Authors:** Robert Kotan, Katalin Peto, Adam Deak, Zsolt Szentkereszty, Norbert Nemeth

**Affiliations:** 1Endocrine Surgery Unit, Linköping University Hospital, Universitetssjukhuset, 581 85 Linköping, Sweden; 2Department of Operative Techniques and Surgical Research, Faculty of Medicine, University of Debrecen, Moricz Zsigmond ut 22, H-4032 Debrecen, Hungary; 3Department of Surgery, Faculty of Medicine, University of Debrecen, Moricz Zsigmond ut 22, H-4032 Debrecen, Hungary

**Keywords:** acute pancreatitis, hemorheology, microcirculation, red blood cell deformability, red blood cell aggregation

## Abstract

Acute pancreatitis still means a serious challenge in clinical practice. Its pathomechanism is complex and has yet to be fully elucidated. Rheological properties of blood play an important role in tissue perfusion and show non-specific changes in acute pancreatitis. An increase in blood and plasma viscosity, impairment of red blood cell deformability, and enhanced red blood cell aggregation caused by metabolic, inflammatory, free radical-related changes and mechanical stress contribute to the deterioration of the blood flow in the large vessels and also in the microcirculation. Revealing the significance of these changes in acute pancreatitis may better explain the pathogenesis and optimize the therapy. In this review, we give an overview of the role of impaired microcirculation by changes in hemorheological properties in acute pancreatitis.

## 1. Introduction

Worldwide, the incidence of acute pancreatitis is 34 per 100,000 people in the general population, and it is rising [1,2,3]. In 17 countries across Europe, it ranges from 4.6 to 100 per 100,000 population, usually higher in Eastern or Northern Europe [4]. Early death cases (within 1–2 weeks) are dominantly caused by the systemic inflammatory response reaction (SIRS) and multiple organ dysfunction syndrome (MODS), late death cases (2–4 weeks or more) are linked to pancreatic necrosis, superinfection of necrosis, sepsis, and MODS [5,6]. Acute pancreatitis research is intensive worldwide, with a growing number of publications per year [7]. 

In the last couple of years, our knowledge of the pathogenesis of acute pancreatitis has significantly diversified. Nevertheless, the medical therapeutic options are still limited, therapeutic success and patient outcomes are poor, especially in the severe form of the disease [3,8,9,10,11,12,13,14,15]. Therefore, an increasing number of papers have been published in the last decades. In the Pubmed database (as searched on 24 November 2022), for the keywords ‘acute pancreatitis’, 75,909 matches could be found, from which 28,778 were published in the last 10 years. The keywords ‘acute pancreatitis pathophysiology’ show 6712 matches (1893 from the last 10 years), while ‘acute pancreatitis microcirculation’ and ‘acute pancreatitis hemorheology’ give 485 (98 in the last 10 years) and 267 (33 in the last 10 years) matches, respectively. 

From these publications, several studies, partly reviewed in this paper, demonstrated that hemorheological and microcirculatory disturbances may play an important role in the complex pathophysiology of acute pancreatitis. These parameters may have therapeutic relevance as well. However, the magnitude of these changes and their interactions have not been fully and adequately studied yet.

## 2. A Brief Overview of the Pathophysiology of Acute Pancreatitis

During the early pathophysiological events, activation of the digestive enzymes and oxygen-free radicals provoke local tissue damage. The release of vasoactive substances and proinflammatory cytokines, which are produced initially in the pancreas and later in the liver, lungs, and spleen, induces systemic inflammatory response syndrome [3,5,16,17]. This can lead to multiorgan failure related to high morbidity and mortality in the severe forms of pancreatitis. 

Independently of the initial noxa, the intra-pancreatic activation of trypsinogen to trypsin is the first event of acute pancreatitis. The central events are the release of proinflammatory mediators (cytokines, vasoactive substances, free oxygen radicals) and the deterioration of microcirculation, and the activation of leukocytes and their rolling and infiltration into the tissue. These mediators are produced in the pancreas in the initial period of pancreatitis but later in the liver, lungs, and spleen as well [3,5,9,18,19,20,21].

Cellular events include pathological calcium signaling, mitochondrial dysfunction, premature trypsinogen activation within the acinar cells and macrophages, endoplasmic reticulum (ER) stress, impaired unfolded protein response (UPR), and impaired autophagy [2,19,22]. The immune response to acinar cell injury is complex. It includes generation of damage-associated molecular patterns (DAMP) leading to monocyte and neutrophil activation. Tumor necrosis factor-α, interleukin-1β, IL-6, IL-18, *monocyte chemoattractant protein-1* (*MCP-1*/CCL2), chemokine (C-X-C motif) ligand 1 and 2 (*CXCL1, CXCL2*), *high mobility group box 1* protein (HMBG1), intercellular adhesion molecule 1 (ICAM-1) entering to the circulation promote chemotaxis and further activation of monocytes and neutrophils, and platelets, leading to local and systemic effects [2,16,19,23,24,25]. Neutrophil extracellular trap formation (NETosis) also has an important role [26]. Circulating histones may also act as potential biomarkers of disease severity [27]. Inflammation, oxidative stress, edema, ductal obstruction, and perfusion disturbance further aggravate acinar cell injury and necrosis [2].

The mechanism for this secondary and systemic proinflammatory cytokines release is unknown, but the injury of the bowel mucosa barrier and the unknown factors involving the mesenteric lymph duct may play a role in the generalized deterioration of microcirculation and SIRS [21,28,29]. The development of microcirculatory disturbances seems to be one of the most important factors in the progression of necrotizing pancreatitis. It has been demonstrated that the impairment of pancreatic microcirculation correlates with the severity of acute pancreatitis [30,31,32,33].

Severe pancreatic ischemia can play a pathogenetic role in the initiation of acute pancreatitis. The platelet adhesiveness and aggregation are increased in the early stages of acute pancreatitis. The connection between platelet activation through reactive oxygen species (ROS) release and the changes in microcirculation have been reported [23,34,35,36,37]. Capillary plugging by leukocytes, especially activated ones, red blood cell deformability impairment, and increased aggregation, together with a decreased functional capillary density, can lead to significant flow changes in the microvasculature [31,38]. High levels of endothelin-1 cause constriction of arterioles and venules, and its antagonist nitric oxide (NO) may have a prognostic and therapeutic role in severe acute pancreatitis [39]. Although acute edematous pancreatitis is characterized by increased and homogeneous microperfusion, experimental necrotizing pancreatitis shows a progressive decrease of capillary perfusion despite stable macro-hemodynamics [31,40,41,42,43]. What is the role of blood flow properties in these processes? How can they contribute to pathophysiological events? Can they serve as therapeutic targets? 

## 3. Relation of Hemorheology and Microcirculation

Rheological properties of the blood play a pivotal role in determining the fluidity of the blood, flow properties, tissue perfusion, and via mechano-sensor pathways, these features are important in modulating numerous endothelial functions [38,44,45,46]. Blood viscosity [mPas], which has non-Newtonian fluid characteristics (as blood viscosity is shear-rate-dependent), is determined mainly by plasma viscosity, hematocrit, red blood cell deformability, and red blood cell aggregation [44,47,48]. Plasma viscosity [mPas] depends on its macromolecule content (primarily fibrinogen, triglycerides, lipoproteins) [47,48]. 

Red blood cell deformability (e.g., elongation index in the function of shear stress measured by ektacytometry method) is determined by the cell morphology and volume, the surface-to-volume ratio, the intracellular viscosity (quantitative and qualitative properties of hemoglobin content), as well as by the viscoelastic properties of the cell membrane and associated structural protein network [47,49]. Red blood cell aggregation (aggregation index and aggregation time dynamic parameters measured by light-transmission or light-reflections methods) depends on plasmatic factors (plasma proteins: Mainly fibrinogen, C-reactive protein, immunoglobulin M) and cellular factors (cell shape and deformability, characteristics of the glycocalyx layer of the cells). The aggregation process is a sensitive and dynamic relation of aggregating forces (weak, not completely understood mechanism by bridging and depletion hypotheses) and disaggregating ones (shear forces, membrane strain, electrostatic repulsion) [50,51]. Related to these determinants, micro-rheological parameters can be influenced by numerous factors, such as alterations in acid-base parameters, pH, osmolarity, free radical reactions, and mechanical trauma. Accordingly, macro- and micro-rheological alterations can be observed in various pathophysiological processes, such as inflammation, sepsis, ischemia-reperfusion, metabolic disturbances, vascular and hematological disorders, among others [44,45,46,48,49,51,52]. 

Increased blood and plasma viscosity, impaired red blood cell deformability, enhanced red blood cell aggregation lead to decreased tissue perfusion [38,44,45,47,48]. Red blood cells with decreased deformability may elevate blood viscosity (so decreasing blood fluidity) and in the microcirculation, it may result in difficulties in passing through microcapillaries [44,48]. Increased red blood cell aggregation leads to an increase in flow resistance as well because of the sum effect of increased particle size in bulk flow, increased disaggregation energy in the microcirculatory bed, decreased frictional resistance at the vessel wall, decreased tissue hematocrit, and increased vascular tone [44,50].

Enhanced aggregation of red blood cells enlarges the circulating particles in the bloodstream. The consequence is a more expressed axial migration of the aggregates along the flow in the vessel, resulting in a widening Poiseuille-zone, which facilitates the margination of leukocytes and platelets and so contributes to their interactions with the endothelium [44,53]. Therefore, deterioration of micro-rheological parameters may appear in inflammatory processes, such as acute pancreatitis. To study these relations and to reveal their significance in pathomechanism and in therapeutic approaches, experimental models are still needed. However, similarities and differences in animal models and human clinical conditions must be taken into consideration when evaluating the results.

## 4. Concerning Animal Models of Acute Pancreatitis

There are several invasive and non-invasive experimental models to investigate the pathogenesis and therapeutic possibilities of acute pancreatitis [54,55,56,57,58,59,60,61,62,63]. Non-invasive methods use hormones and various chemicals such as cerulein, anti-cholinesterase insecticide, ethyl-alcohol, ovalbumin, various diets (e.g., a choline-deficient diet with ethionine), L-arginine, L-ornithine, L-lysine, L-histidine, immune-mediated models and gene knockout animals (e.g., interleukin-1, IL-6, IL-10, tumor necrosis factor-α, chemoattractant cytokine receptor-1, neurokinin-1 receptor, intercellular adhesion molecule 1, cathepsin B, granulocyte-macrophage colony-stimulating factors knockouts). Invasive methods use a closed duodenal loop, antegrade pancreatic duct perfusion, biliopancreatic duct injection, and a combination of secretory hyperstimulation with minimal intraductal bile acid exposure, vascular-induced, ischemia-reperfusion (oxidative stress) and duct ligation, endoscopic retrograde cholangiopancreatography (ERCP), and combined methods [55,57,59,61,63]. The benefits of invasive models are their reproducibility, the non-invasive models are cheap and simple. The experiments on rodents are the cheapest, therefore, they are suitable for collecting a large amount of data, but they make the vital measurements difficult, and they cannot properly reflect the real clinical situation [55,59,63].

If we focus on the experimental studies, we can reveal that there are two main types of experimental work related to the timing of the therapy of acute pancreatitis. Numerous studies focused on the first several hours after the onset of disease, however, the clinical relevance is low since more often, patients are hospitalized in a latter, symptomatic stage. Most of the early-type models help, first of all, to better understand the pathogenesis of acute pancreatitis, while the models with advanced stage of the disorder may provide more realistic condition to investigate therapeutic approaches [55,57,59,60,63].

Further challenges of animal models include anatomical and physiological inter-species differences [54,57,58,64,65,66,67,68,69,70]. The majority of the animal models are conducted on rodents, dominantly in mice and rats [70]. In mice, the pancreas structure anatomically is diffuse/dendritic, lobular, and soft, the main duct is connected to the bile duct proximally to its entry to the duodenum, and there are numerous accessory ducts. The diameter of the lobules is 0.5–1.5 mm (vs. human 1–10 mm), the number of islets is 1000–5000 (vs. human 1,000,000–15,000,000), and their location is more random and interlobular [69,70]. Concerning microvasculature, in mice insulo-venous system prevails, and insulo-acinar portal system exists (in humans insulo-acinar portal system prevails) [65,70]. The dominant perfusion order is center-to-periphery (2/3) and polar (1/3), while in humans, it is most likely polar. Content ratios of β and α cells are comparable (β cells: 50–70% in humans and 60–80% in mice, α cells: 20–40% in humans and 10–20% in mice). Furthermore, in humans, the sympathetic and parasympathetic innervations are richer [68,70]. These characteristics are important for planning and performing microcirculatory investigations in acute pancreatitis models and for the extrapolation of the results.

## 5. Altered Hemorheological Factors Influencing Microcirculation in Acute Pancreatitis 

Despite the large amount of clinical and experimental studies on acute pancreatitis, hemorheological and microcirculatory aspects of the disease have yet to be fully elucidated. In acute pancreatitis, all the major hemorheological parameters may worsen: Increase of blood and plasma viscosity, impairment in red blood cell deformability, and enhanced red blood cell aggregation (Table 1). 

Hemoconcentration by dehydration is a significant and early pathophysiological change in acute pancreatitis [41,71,72,73]. In the past decades, there were several studies that tested the role of hematocrit as a severity predictor of acute pancreatitis [71,73]. The importance of the elevated level of hematocrit at admission has been known for over 50 years. The aim of earlier and current studies is to find a connection between the admission of hematocrit (within the initial 24 h of hospitalization) and necrotizing pancreatitis and organ failure. It has been revealed that early hemoconcentration is a useful marker to predict the severity of acute pancreatitis but not every study could demonstrate it. The cause of the difficulties may originate from the type and sample size of the study, the cut-off values used, and the treatment [71,72,73].

In acute pancreatitis, the inflammatory processes, free radical reactions, complement activation, as well as metabolic changes, and alteration in acid-base parameters and pH, all may lead to micro-rheological alterations (Figure 1).

Earlier, the role of reactive oxygen species (ROS) in the mechanism of the pathogenesis of acute pancreatitis was revealed [23,34,35,36,37,81]. Red blood cells are very vulnerable to oxidative stress and free radical reactions, as they are rich in iron cation (a catalyst in Fenton-reaction), and these cells, without nucleus, cannot repair/replace their damaged cellular components. Oxygen-centered free radicals in inflammatory and ischemia-reperfusion processes may damage red blood cells’ membrane (by lipidperoxidation), protein structure (by crosslinking the sulfhydryl chains), and hemoglobin content (by forming methemoglobin and Heinz-bodies) [45,48]. A decrease in pH (by an increase of H^+^ and lactate concentration) transforms the normal biconcave discocyte shape of the erythrocytes toward stomatocyte and sphero-stomatocyte morphology, while ATP depletion, ROS-derived damage, and calcium accumulation turn the cells to echinocyte and sphero-echinocyte morphology [45,48,49,82,83]. Both are associated with decreased deformability. Changes in osmolarity (by altering the volume and density of cells) and disturbance in oxygenation also result in impaired red blood cell deformability, affecting their aggregability as well (e.g., deoxygenated red blood cells express enhanced aggregation) [48,50]. Mechanical trauma to blood (higher than physiological shearing and compressing forces) can lead to sublethal red blood cell trauma with decreased deformability and enhanced aggregation, microvesicle generation, and mechanical hemolysis, depending on the magnitude and exposure time of the mechanical stress [45,48,84].

The platelets with activation are able to release ROS, nitric oxide, and proinflammatory mediators [23,24]. The imbalance of nitric oxide and endothelin with the predominance of the endothelin effect leads to vasoconstriction and pancreatic edema due to increased vascular permeability [33,34,85]. Platelet activation is accompanied by leukocyte activation, and they infiltrate into the pericapillary tissue leading to further tissue damage. The leukocyte adhesion through endothelial cells (integrins, ICAM-1) can lead to higher interaction with vascular function, the increased capillary permeability causes a massive outflow of plasma, increased blood viscosity can reduce capillary blood flow velocity [33]. Experimental necrotizing pancreatitis is characterized by decreased pancreatic microcirculation despite stable macro-hemodynamics. This phenomenon, leukocyte transmigration and release of protease and free radicals, plays a role in the injury of the intestinal mucosa barrier leading to bacterial translocation leading to superinfection of sterile necrosis, multiple organ failure, and worsening of inflammatory cascade [2,3,29,86].

Gut injury during inflammation associated with bacterial translocation can play a significant role in the mechanism of superinfection of the pancreas and peripancreatic necrosis, the acute peripancreatic fluid collection, the post-necrotic fluid collection, and the walled-of pancreatic necrosis, as well as in the multiple organ failure in acute pancreatitis [29]. Bacterial superinfection in acute pancreatitis is originated from the gut system (the most common bacterium cultures from the local septic complications in acute pancreatitis are enteral bacteria). Although these local complications present mostly in the later phase of pancreatitis (later than 2–3 weeks after the onset of the disease) to save the gut barrier function has a significant role in the treatment of acute pancreatitis. Thus, early enteral feeding, epidural anesthesia, the prevention and treatment of intraabdominal hypertension are routinely used methods. Ameliorating the microcirculation of the gut may play an important role in preserving the gut barrier function and to avoid bacterial superinfection.

Zaets et al. demonstrated in the trauma hemorrhagic-shock model that gut injury and gut-derived factors carried in the intestinal lymph can influence red blood cell deformability [87]. However, the hemorrhagic shock itself causes red blood cell damage [88,89]. Experimental data also showed that mesenteric lymph duct ligation can decrease the impairment in red blood cell deformability in acute pancreatitis [28], as mesenteric lymph in acute pancreatitis contains factors causing red blood cell damage [28,90]. The role of biologically active mesenteric lymph in the pathogenesis of multiorgan failure in critically ill patients, such as in acute pancreatitis, is widely investigated, but further research is necessary. 

Shanbhag et al. could provoke cardiac dysfunction in rats with acute pancreatitis-conditioned mesenteric lymph, which enhances the “gut-lymph” hypothesis in acute pancreatitis. It was demonstrated that the cardiac dysfunction with evidence of increased inflammation and tissue injury was not associated with low blood pressure but with mesenteric lymph toxicity. The analysis of the molecular composition of mesenteric lymph in acute pancreatitis showed increasing pancreatic catabolic enzymes [91].

In severe cases, the appearing sepsis [92,93,94,95] and abdominal compartment syndrome [96], by their nature and by similar non-specific mechanisms, may further worsen the micro-rheological and microcirculatory parameters. Increased fibrinogen and C-reactive protein concentration can enhance red blood cell aggregation [48,97,98,99]. Vanderelst et al. observed a rapid decrease in the expression of CD35 membrane protein (protects against complement activation) in red blood cells in septic patients. This alteration with decreased red blood cell deformability and shape alterations can facilitate erythrophagocytosis [100].

Acute pancreatitis impairs pancreatic and systemic microcirculation [30,31,33,79,80]. Microcirculatory disturbances in acute pancreatitis include vasoconstriction, shunting, progressive exclusion of capillaries, edema formation, neutrophil adhesion, micro-thrombus formation, hemorrhage, obstructed lymph flow, increased blood viscosity, decreased red blood cell deformability, and enhanced aggregation [33,90]. These alterations may contribute to the development of the widely studied ‘no-reflow’ microcirculatory phenomenon [101,102,103].

## 6. Therapeutic Approaches Related to Hemorheology and Microcirculation

Effective preventive and therapeutic drugs for the human acute pancreatitis are still lacking in the mirror of the evaluation of prognostic factors [12,83,104,105,106,107,108]. Although they are non-specific, the early hydration and enteral feeding are still important tools [11,12,109,110]. Both basic and clinical studies have focused on the means of preventing or ameliorating the damaging effects of microcirculatory and/or hemorheological disorders [79,111]. Most of the monotherapies have an effect in the early period of acute pancreatitis and can be useful in ERCP-related pancreatitis or in the case of pancreas transplantation. The therapy of acute pancreatitis is still symptomatic despite numerous promising experimental research. Early and adequate fluid resuscitation decreases the harmful effect of systemic inflammatory response syndrome, organ failure, and mortality, according to recent clinical guidelines and studies [11,12,110]. Different monotherapies attempt to alleviate or abort the inflammatory cascade in the severe form of the disease at various attack points.

As discussed earlier, the development of acute pancreatitis begins with pancreas protease activation and contribution of numerous factors—leukocyte activation and rolling, free radicals, vasoactive mediators, cytokines, endothel activation—lead to microcirculatory disorders and inflammatory cascade in the pancreas and in different organs, causing severe pancreatitis with systemic complication. Thus, immunomodulatory therapeutic strategies also have great importance [107,112]. Several studies revealed a positive, protective effect of prophylactic therapy in the very early phase of acute pancreatitis, for example the prophylactic anti-endothelin or NO therapy. The role of gut mucosal barrier injury and mesenteric lymph in acute pancreatitis and inflammatory cascade is described earlier [28,29]. Knowledge of the early phase of pancreatitis is very important, but later, the process becomes more complex.

There are only few data about pharmaceutical agents that modulate rheological factors. In experimental, cerulein-induced acute pancreatitis, an improvement of microcirculation and micro-rheological parameters (red blood cell deformability and aggregation) was found using low molecular weight heparin (enoxaparine) or flunixin (a non-steroid 7nti-inflammatory agent) administration [79,113]. Surprisingly, the improving effect of pentoxifylline on micro-rheological parameters was not obvious (red blood cell deformability did not improve better than in the other treated groups), however, microcirculatory parameters improved [79,114]. The next step should be to find the optimal dosage (prophylactic or higher dosage) of enoxaparine or other LMWH drugs in the early phase of acute pancreatitis. The early correct fluid resuscitation in acute pancreatitis is well-known, however, it may contribute to the increasing intraabdominal pressure in case of over-dosage of fluid intake. Therefore, observation of intraabdominal pressure is a useful method to avoid intraabdominal hypertension or abdominal compartment syndrome [96].

The multifactorial pathogenesis and the moderate success of monotherapies in experimental studies, and the almost unsuccessful application of these drugs in the clinical routine lead logically to the idea that multidrug therapy may have more effect. Werner et al. reported promising results with multidrug therapies in an experimental study [115]. The human application of multidrug therapies is still waiting for further studies.

## 7. Conclusions

The investigation of the hemorheology and microcirculation in acute pancreatitis is an old-new area of research and it may help to better explain the pathogenesis and optimize the therapy. Rheological (macro and micro) alterations of blood play an important role in the pathophysiology of acute pancreatitis. Altered blood rheology may cause further deterioration in the microcirculation, contributing to the progression of the disease with local and systemic effects. The optimal treatment of severe acute pancreatitis should also focus more on the improvement of microcirculation in order to prevent the damage of gut barrier function and enteral-origin superinfection as well.

Various large and small animal models are still important to better understand the pathophysiology of acute pancreatitis and reveal further therapeutic targets. However, clinically more relevant models are still needed to define the optimal therapeutical strategy (e.g., drugs, dosage, timing).

## Figures and Tables

**Figure 1 metabolites-13-00004-f001:**
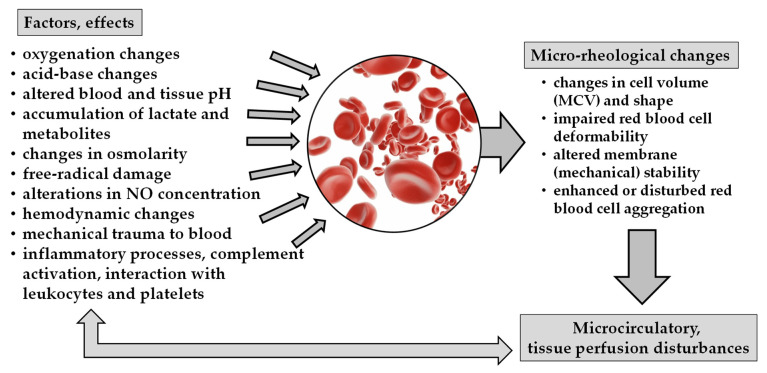
Schematic overview of factors and effects causing micro-rheological changes and their consequences on microcirculation.

**Table 1 metabolites-13-00004-t001:** Summary of main changes of hemorheological and microcirculatory factors in acute pancreatitis.

Parameter/Variable	Main Changes	Reference Example	
		Clinical	Experimental
Blood viscosity	Increase (mainly by hemoconcentration, dehydration)	[71,72,73]	[41]
Plasma viscosity	Increase (depending on fibrinogen, lipoproteins, and triglyceride concentration)	[74,75]	[41]
Red blood cell deformability (e.g., elongation index)	Impairment *	[35,76]	[28,77,78,79]
Red blood cell aggregation (e.g., aggregation index)	Enhancement *	[74]	[41,79]
Microcirculatory parameters (e.g., blood flux unit, perfusion units, functional capillary density)	General deterioration *	[32,42,80]	[18,30,31,32,41,43]

***** Main causes are summarized in Figure 1.

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
