# Peer review of "Hemorheological and Microcirculatory Relations of Acute Pancreatitis"

_metabolites, 2022, doi:10.3390/metabo13010004_

Round 1

Reviewer 1 Report

It is an extensive narrative review. There are a few points:

1. Section 1 can be significantly shortened and should lead to Section 2.

2. Section 2 does not mention the quantitative parameters for rheological factors. If it covers it in this segment, the authors do not have to repeat it later.

3. Section 3 has far too many details related to models that are not relevant to this review. That can be separate review unto itself.

4. This is the most significant section and should be put into perspective and centre of the manuscript. What are the positives and negatives of each parameter? Are there confounders like dehydration for example? How can one correct for these confounders in the quantitative parameters? It is not enough to write that 'the background of these changes is complex'. Which of the quantitative parameters are less affected by confounders? Are there any surrogates for quantitation?

5. Figure 1 needs editing and formatting. Explain in the legend why the dotted arrow is double ended?

6. Lines 198 to 230 should have been summarised and added to Section 1.

7. What polypharmacy do the authors envisage to modulate rheological factors? Is there a magic bullet? Lines 283-287 do not make sense... Please rephrase.

8. What do the authors suggest as a concrete step forward? This should be included in the conclusions.

Author Response

Dear Reviewer,

thank you very much for your time and for your valuable comments. Accordingly, the necessary revision and clarifications, including some re-compositions and extension of certain part have been made. All the corrections and changes are in red color in the revised manuscript.

  1. The Introduction (without pathophysiology) has been separated and revised. In a separate chapter (as a new chapter 2) the brief pathophysiology overview has been clarified, leading more concisely to the next part. “…Therefore, an intensively increasing number of papers have been published in the last decades. In the Pubmed database (as searched on 24 November, 2022) for the keywords ‘acute pancreatitis’ 75.909 matches could be found, from which 28.778 have been published in the last 10 years. The keywords ‘acute pancreatitis pathophysiology’ show 6.712 matches (1.893 from the last 10 years), while ‘acute pancreatitis microcirculation’ and ‘acute pancreatitis hemorheology’ give 485 (98 in the last 10 years) and 267 (33 in the last 10 years) matches, respectively. From these publications there are several studies –partly overviewed in this paper– demonstrated that hemorheological and microcirculatory disturbances may play an important role in the complex pathophysiology of acute pancreatitis. These parameters may have therapeutic relevance as well. However, the magnitude of these changes and their interactions have not been fully elucidated yet.”

  1. The hemorheology&microcirculation chapter (explaining better the measurable parameters as suggested) have been clarified and extended: Red blood cell deformability (e.g., elongation index in the function of shear stress measured by ektacytometry method)…”, „Red blood cell aggregation (aggregation index and aggregation time dynamic parameters measured by light-transmission or light-reflections methods) depends on…”, „Therefore, deterioration of micro-rheological parameters may appear in inflammatory processes, such as acute pancreatitis. To study these relations and to reveal their significance in the pathomechanism and possibly in therapeutic approaches, experimental models are still need. However, similarities and differences of animal models and human clinical condition must be taken into consideration when evaluating the results.”

  1. Besides necessary corrections, clarifying sentences on the need of overviewing the differences of pancreas anatomy (especially on the vasculature) had been included, e.g. These characteristics are important for planning and performing microcirculatory investigations in acute pancreatitis models and for extrapolation of the results.”

  1. The explanation of changes of micro-rheological parameters (dehydration, free radical, metabolic alteration, etc., and their interactions) have been described in more details and with further references. Red blood cells are very vulnerable to oxidative stress and free radical reactions, as they are rich in iron cation (a catalyst in Fenton-reaction) and these cells without nucleus cannot repair/replace their damaged cellular components. Oxygen-centered free radicals in inflammatory and ischemia-reperfusion processes may damage red blood cells’ membrane (by lipidperoxidation), protein structure (by crosslinking the sulfhydryl chains) and hemoglobin content (by forming methemoglonin and Heinz-bodies) [45,48]. Decrease in pH (by increase of H+ and lactate concentration) transforms the normal biconcave discocyte shape of the erythrocytes toward stomatocyte and sphero-stomatocyte morphology, while ATP depletion, ROS-derived damage and calcium accumulation turn the cells to echinocyte and sphero-echinocyte morphology [45,48,49,82,83]. Both morphological directions are associated with decreased deformability. Changes in osmolarity (by altering volume and density of cells) and disturbance in oxygenation also result in impaired red blood cell deformability, affecting their aggregability as well (e.g., deoxygenated red blood cells express enhanced aggregation) [48,50]. Mechanical trauma to blood (higher than physiological shearing and compessing forces) can lead to sublethal red blood cell trauma with decreased deformability and enhanced aggregation, microvesicle generation and mechanical hemolysis, depending on the magnitude and exposure time of the mechanical stress [45,48,84].”

  1. Figure 1 has been corrected, Table 1 has been clarified (as requested in the Reviewer’s notes in pdf-review file) with concrete parameters, such as red blood cell deformability (e.g., elongation index), red blood cell aggregation (e.g., aggregation index), microcirculatory parameters (e.g, blood flux unit, perfusion units, functional capillary density), based on the references.

  1. Several sentences have been corrected, and by the rewritten, recomposed chapter the indicated sentences in lines 198-230 had been clarified.

  1. Unfortunately, there is no “magic bullet” in hemorheological therapy, therefore the known possibilities could be mentioned in the related chapter. Here we clarified some editing errors as well.

  1. As a concrete step forward we added these sentences in the therapy chapter: “The next step should be to find the optimal dosage (prophylactic or higher dosage) of enoxaparine or other LMWH drugs in the early phase of acute pancreatitis. The early correct fluid resuscitation in acute pancreatitis is well-known, however, it may con-tribute to the increasing intraabdominal pressure in case of over-dosage of fluid intake. Therefore, observation of intraabdominal pressure is a useful method to avoid intraabdominal hypertension or abdominal compartment syndrome [96].”, and in the Conclusions: “The optimal treatment of severe acute pancreatitis should also focus better on the improvement of microcirculation in order to prevent the damage of gut barrier function and enteral-origin superinfection as well.”

Thank you very much again for the valuable review. We hope that the corrections in the revised version might be acceptable.

Sincerely yours,

Norbert Nemeth

Reviewer 2 Report

Reviewer Comments to the Author

In this study, the authors gives an overview of the role of impaired microcirculation by changes in hemorheological properties in acute pancreatitis. Overall, the article has substantial content and reasonable structure, however, there are some problems to be further improved as well:

1. In the introduction section, the author describes pathophysiological events of Acute Pancreatitis in paragraphs 3,4, and 5 ofcreatitis, but we focus more on hemorheological and microcirculatory relations of it. So more content should be introduced from these aspects.

2. In part 4, Gut injury during inflammati However, the hemorrhagic....her researchis necessary” and "Shanbhag et al. could provoke cardiac ...... eight pancreatic catabolic enzymes [88]. The two paragraphs may be merged.

3.  DeleteFurther analysis of factors demonstrated that kynurenine and specific miRNA may have a prognostic role in the evaluation of the disease severity and organ failure [87].”

4. What kind of role did gut injury play in the mechanism of acute pancreatitis? This may be a good point of study and authors can focus more on research in this area.

Author Response

Dear Reviewer,

thank you very much for your time and for your valuable comments. Accordingly, the necessary revision and clarifications, including some re-compositions and extension of certain part have been made.

The Introduction (without pathophysiology) has been separated and revised. In a separate chapter the brief pathophysiology overview has been clarified, as well as the hemorheology&microcirculation chapter. Clarifying sentences on the need of overviewing the differences of pancreas anatomy (especially on the vasculature) had been included. The explanation of changes of micro-rheological parameters (dehydration, free radical, metabolic alteration, etc., and their interactions) have been described in more details and with further references. More details about the mechanism and significance of gut injury have been included. The suggested modifications with deleting some confusing sentences have been also completed.

Thank you very much again for the valuable review. We hope that the corrections in the revised version might be acceptable.

Sincerely yours,
Norbert Nemeth

Round 2

Reviewer 1 Report

I have added comments directly to the PDF. Please check.
Best wishes.

Author Response

Dear Reviewer,

many thanks for the careful review and the corrections indicated in the review-pdf. Accordingly, all the misspellings, typos and rewordings had been corrected or made in the second revised version (pages 1, 2, 5, 6).

Thank you!